# The Intriguing Biogeographic Pattern of the Italian Wall Lizard *Podarcis siculus* (Squamata: Lacertidae) in the Tuscan Archipelago Reveals the Existence of a New Ancient Insular Clade

**DOI:** 10.3390/ani13030386

**Published:** 2023-01-23

**Authors:** Francesco Gallozzi, Claudia Corti, Riccardo Castiglia, Vasco Avramo, Gabriele Senczuk, Claudia Mattioni, Paolo Colangelo

**Affiliations:** 1Department of Biology and Biotechnologies ‘Charles Darwin’, Sapienza University of Rome, Via Borelli 50, 00188 Rome, Italy; 2National Research Council, Research Institute on Terrestrial Ecosystems, Via Salaria km 29.300, 00015 Rome, Italy; 3Museum of Natural History ‘La Specola’, Via Romana 17, 50125 Florence, Italy; 4Institute for Environmental Protection and Research (ISPRA), Via Cà Fornacetta, 9, Ozzano Emilia, 40064 Bologna, Italy; 5Department of Agricultural, Environmental and Food Sciences, University of Molise, 86100 Campobasso, Italy; 6National Biodiversity Future Center, 90133 Palermo, Italy

**Keywords:** *Podarcis siculus*, insular lizards, biogeography, Mediterranean, Tuscan Archipelago

## Abstract

**Simple Summary:**

Islands represent hotspots of biodiversity and ideal places to study evolutionary processes. Wall lizards of the genus *Podarcis* are widespread and well-studied species in Europe, and they are present on several islands and islets across the Mediterranean. They show a remarkable hidden insular biodiversity that has often revealed the presence of undescribed biological entities. In this study, both the genetic and morphological diversity of the populations of *P. siculus* of the Tuscan Archipelago were investigated. Our results revealed an intriguing biogeographical pattern which could be the result of at least two different colonization waves. Particularly, we identified a new ancient insular clade from Giglio and Capraia islands which surely deserves conservation efforts and further taxonomic investigation.

**Abstract:**

The Tuscan Archipelago is one of the most ancient and ecologically heterogeneous island systems in the Mediterranean. The biodiversity of these islands was strongly shaped by the Pliocene and Pleistocene sea regressions and transgression, resulting in different waves of colonization and isolation of species coming from the mainland. The Italian wall lizard, *Podarcis siculus*, is present on the following islands of the Tuscan Archipelago: Elba, Giglio, Giannutri, Capraia, Montecristo and Cerboli. The species in the area displays a relatively high morphological variability that in the past led to the description of several subspecies. In this study, both the genetic and morphological diversity of *P. siculus* of the Tuscan Archipelago were investigated. Specifically, the meristic characters and the dorsal pattern were analyzed, while the genetic relationships among these populations were explored with mtDNA and microsatellite nuclear markers to reconstruct the colonization history of the Archipelago. Our results converge in the identification of at least two different waves of colonization in the Archipelago: Elba, and the populations of Cerboli and Montecristo probably originate from historical introductions from mainland Tuscany, while those of Giglio and Capraia are surviving populations of an ancient lineage which colonized the Tuscan Archipelago during the Pliocene and which shares a common ancestry with the *P. siculus* populations of south-eastern Italy. Giannutri perhaps represents an interesting case of hybridization between the populations from mainland Tuscany and the Giglio-Capraia clade. Based on the high phenotypic and molecular distinctiveness of this ancient clade, these populations should be treated as distinct units deserving conservation and management efforts as well as further investigation to assess their taxonomic status.

## 1. Introduction

Island systems are recognized as hotspots of biodiversity across the world, and they can be seen as a real cauldron of evolution. The long-time isolation and the peculiar environmental conditions can trigger deep intraspecific and interspecific biological diversification, especially in small islands [1]. The Tuscan Archipelago is one of the most ancient and ecologically heterogeneous island systems in the Mediterranean Sea. It is composed of seven main islands and several islets lying between Corsica and the Italian Peninsula with a great diversity of biotopes (from deciduous Mediterranean oak forest to bush-shrub lands). The biodiversity of these islands was strongly shaped by the Pliocene and Pleistocene sea regressions and transgression [2,3,4]. This phenomenon is well known as a source of biological evolution, allowing waves of colonization of species from the mainland during periods of glacial marine regression, followed by isolation due to the rising sea levels. In this context, many different organisms were able to evolve, diversify, and become extinct [5,6,7,8]. While some islands of the Tuscan Archipelago were connected to the mainland during the Pleistocene (land bridge islands), others remained isolated. For example, the low sea level during the last glacial maximum probably led Elba, Pianosa and Giannutri to be connected with mainland Tuscany, while Montecristo, Capraia, Gorgona and Giglio Islands remained isolated (Figure 1).

The islands of the Tuscan Archipelago host a rich and peculiar biological community, strongly influenced by historical, biological and current eco-geographical factors [2,4,9]. Furthermore, since ancient times, humans have favored the spread of species between the various islands and between them and the continent by altering the community composition [10,11,12,13]. Some examples are the presence of *Vipera aspis hugyi* on Montecristo or *Zamenis longissimus* on Elba [14,15]. All these factors contribute to the islands’ complex and diverse biological community, which in many cases has remained little studied or completely unknown.

The wall lizards of the genus *Podarcis* are widespread and well-studied species in Europe, and they are present on several islands and islets across the Mediterranean. The remarkable hidden insular biodiversity of this group in terms of deep genetic and phenotypic divergence has often revealed the presence of unique biological entities that may be important with regard to their conservation [16,17,18,19,20,21,22,23,24,25,26,27,28]. *Podarcis siculus* is present on the following islands of the Tuscan Archipelago: Elba, Giglio, Giannutri, Capraia, Montecristo and Cerboli. The species displays in the area a relatively high morphological variability that in the past led to the description of several subspecies [29]. Despite this, only one study has been conducted on the genetic variability of *P. siculus* of the Tuscan Archipelago [17] using allozymes, while no molecular analysis based on mtDNA and nuclear markers is available yet.

In this study, both the genetic and morphological diversity of *P. siculus* in the Tuscan Archipelago were investigated in order to shed new light on the history of the colonization of the Archipelago by this species. Specifically, the meristic characters and the dorsal pattern of a large number of specimens of the islands of Elba, Giglio, Giannutri, Capraia, Cerboli and Montecristo were analyzed. In addition, the genetic relationships among these populations were explored with mtDNA and microsatellite nuclear markers to reconstruct the colonization history of the Archipelago.

## 2. Materials and Methods

### 2.1. Dorsal Pattern Diversity

Differences in dorsal pattern (DP) were studied on 630 adult specimens (393 males and 237 females; details in Table 1) kept at the “La Specola” Museum. Specimens belonging to both mainland Tuscany and six different islands of the Tuscan Archipelago (Giglio, Capraia, Giannutri, Montecristo, Cerboli, Elba) were analyzed. Each specimen was assigned to one of the six defined categories of DPs as shown in Figure 2. The categories are named as follows: “campestris”, showing a brown and/or black continuous or partially interrupted occipital stripe bordered on both sides by a relatively large brown or green parietal band, at least in the posterior half of the body; “calabresiae” (from the name of the subspecies described for Montecristo Island—*P. s. calabresiae* [30]), similar to “campestris” but with very thin stripe-like parietal bands; “reticulated”, showing a complete or partially complete coalescence of the elements composing the occipital stripe and the ones composing the parietal bands; “concolor”, showing no DP; “intermediate 1”, an intermediate phenotype between “campestris” or “calabresiae” and “reticulated”; “intermediate 2”, an intermediate phenotype between “campestris” or “calabresiae” or “reticulated” and “concolor”. To test possible association between phenotypes and islands, a χ^2^ test was conducted separately for males and females.

### 2.2. Morphological Analysis

A subset of 237 adult specimens (151 males, 86 females; see Table 1 for details) belonging to six islands of the Tuscan Archipelago (Giglio, Capraia, Giannutri, Montecristo, Cerboli, Elba) and mainland Tuscany, preserved at the “La Specola” Museum of the Natural History Museum of the University of Florence (MZUF), were morphologically analyzed. The snout-vent length (SVL) was measured at the nearest 0.1 cm and the following meristic characters were counted: (a) number of midbody scales, (b) ventral scales, (c) collar scales, (d) gular scales, (e) femoral pores and (f) subdigital lamellae on the right leg, (g) supraciliary scales, (h) supraciliary granules, (i) supratemporal scales (Appendix A). We then performed a PCA on meristic characters to identify the groups that better explained the morphological diversity in the Archipelago. Successively, we tested the presence of statistically significant differences between sexes and between islands performing an ANOVA on the SVL, and a MANOVA on the meristic data.

### 2.3. Genetic Analysis

29 tissue samples were collected from 2014 to 2021 on Giglio, Capraia, Giannutri, Montecristo, Cerboli and Elba islands. Tissues were obtained from small tail tips by inducing autotomy after light pressure, all the individuals were released at the capture site. Tail tissues were stored in 96% pure ethanol. Genomic DNA was extracted from all the tissue samples by means of the universal extraction protocol [32]. We analyzed the mitochondrial gene cytochrome b (cytb) and eight microsatellite loci. Cytb was amplified using the primer employed by Podnar et al. [33] Cyt F (5′-TTTGGATCACTATTRGGCCTCTGCC-3′) and H15425 (5′-GGTTTACAAGACCAGTGCTTT-3′). Amplification was carried out using the following protocol: an initial denaturation at 94 °C for 2 min followed by 35 cycles with denaturation at 95 °C for 10 s, annealing at 55 °C for 20 s, and elongation at 72 °C for 90 s and a final extension at 72 °C for 7 min. Microsatellite loci were selected from among those available in literature [34,35,36,37] and amplified in three different mixes (mix 1: Pb73, C9; mix 2: Pli24, Pli4, Pli18, Pb10; mix 3: Lv-4-a, Lv-4-19; see Appendix A for details). Amplifications were carried out using different protocols for mix 1–2 and mix 3. For mix 1 and 2, PCR was performed with an initial denaturation step at 94 °C for 5 min, followed by 32 cycles composed of a denaturation at 95 °C for 30 s, annealing at 57°C for 90 s, elongation at 72 °C for 30 s and a final extension at 60 °C for 30 min. For mix 3, PCR was performed with an initial denaturation at 94 °C for 5 min followed by 32 cycles of denaturation at 95 °C for 30 s, annealing at 59 °C for 90 s, elongation at 72 °C for 30 s and a final extension at 60 °C for 30 min. Each individual was genotyped with the ABI3130 system and results were visualized with Thermo Fisher Peak Scanner Software.

The obtained cytb dataset was completed by adding 340 sequences from those available in GenBank from previous studies [25,33] covering the entire distribution area of *Podarcis siculus* (i.e., continental Italy, Croatia, Sicily, Sardinia, Corsica and some smaller islands). No sequences from the Tuscan Archipelago were available in GenBank. Sequences of cytb were aligned with MUSCLE [38] in MEGA (version 11.0.13; [39]). We then built a median joining network using the software PopArt (version 1.7; [40,41]). Successively, we calculated p-distances among the clades/lineages retrieved with the network. Population structure based on microsatellite loci was reconstructed using the spatial PCA (sPCA) as implemented in the R package adegenet (version 2.1.5; [42]). In addition, the genetic clustering was investigated using an admixture analysis with k values from 1 to 4 and the cross-validation score for each k value was calculated. To do that, we used the tess3r package (version 1.1.0; [43]) in R.

## 3. Results

### 3.1. Dorsal Pattern

Considering males only, there is a high frequency of the “calabresiae” phenotype in Montecristo, while mainland Tuscany, Elba and Cerboli islands show a high frequency of the “campestris” phenotype. In the other islands there is a higher diversity of patterns with “reticulated”, and “concolor” phenotypes together with their intermediates. Females follow a similar pattern with the notable difference consisting of the fact that almost all the females of Capraia show the “campestris” phenotype. The χ^2^ test confirmed the statistical significance of the differences in the phenotypic frequencies reported above (*p*-value < 0.05) in males and females. DP frequencies are shown in Figure 3 and in Appendix A.

### 3.2. Morphological Analysis

The PCA results are slightly different between sexes. In males, there are two very distinct groups: one being composed of Giglio, Capraia and Giannutri and one of Elba, Cerboli, Montecristo and mainland Tuscany. In females, we can still distinguish two groups but less distinct than in males: one that includes Giglio and Capraia which are quite similar to each other, and one that includes Elba, Montecristo, Giannutri and mainland Tuscany with Cerboli being intermediate between the groups (Figure 4). MANOVA confirms the significance of these differences in morphological traits between sexes and between localities. On the contrary, ANOVA shows the presence of statistically significant differences in SVL between sexes only, with males being larger than females. There are not statistically significant differences in SVL between the different sampling localities (between islands and between islands and mainland Tuscany). MANOVA and ANOVA results are reported in Table 2 and Table 3, respectively. SVLs are shown in Figure 5.

### 3.3. Genetic Analysis

We successfully obtained 24 cytb sequences from our samples. Elba island was excluded from the analysis due to the very low quality of its cytb sequences. The median-joining network (Figure 6) revealed the presence of the main mitochondrial clades already described by Senczuk et al. [25]: a Siculo–Calabrian lineage and a central–northern lineage splitting into Adriatic and Tyrrhenian clades. Specimens from Cerboli, Giannutri and Montecristo belong to the Tyrrhenian clade. Specifically, four haplotypes are found on these islands: two from Montecristo, one from Giannutri and one from Cerboli. Furthermore, a new undescribed clade, well differentiated from the previous ones, emerged from the analysis. This clade includes all specimens from the islands of Giglio and Capraia. The newly identified Giglio–Capraia clade has eight unique haplotypes found nowhere else in the range of *P. siculus*. These haplotypes appear to be very close to each other and related to the Adriatic clade. It is worth noting that Giglio and Capraia also show a high genetic distance (p-distance) from the mainland populations of *P. siculus* (7.3–7.5% from the Adriatic clade and 8.7–9.1% from the Tyrrhenian clade; Table 4) and that there is also some divergence between the haplotypes of the two islands (5.4%).

All 29 sampled individuals were successfully genotyped using our microsatellites panel (see Appendix A for details). However, only seven out of the eight loci were successfully obtained. In the sPCA (Figure 7), all the islands appear distinct from each other with the sole exception of Elba and Cerboli, which show full overlap.

The results of the admixture analysis are shown in Figure 8. For k = 2 Giglio and Montecristo show no signs of admixture with each other or with the other islands. Conversely, Capraia, Cerboli, Elba and Giannutri, show a signature of admixture of the two genetic clusters. For k = 3, the specimens from Giglio and Capraia, form two separate genetic clusters. The third cluster includes all the Montecristo specimens. The specimens of Cerboli, Elba and Giannutri show signs of admixture between Giglio and Montecristo clusters. For k = 4, Giglio, Capraia and Montecristo keep on being well distinct, while Elba, Cerboli and Giannutri show admixture between clusters. Particularly, Giannutri shows signs of admixture with Giglio. According to cross-validation scores, k = 4 is the best fitting k value (Figure 9).

## 4. Discussion

The final picture that emerges from our results describes a very complex and intriguing biogeographic pattern with a high morphological, phenotypic, and genetic diversity between islands that certainly deserves further investigation.

The previous allozyme analysis by Corti et al. [17] showed that the populations of Giglio, Capraia and Giannutri differ from those of the other islands of the Archipelago which are genetically close to those of the mainland. The morphological distinctiveness of the lizards of Giglio, Giannutri and Capraia compared to those of the remaining islands had already been observed by Mertens [44] and Corti [16]. Mertens defined the lizards of these islands as a race of *Lacerta sicula* joining them under the name *L. s. tyrrhenica*, thus highlighting the distinctiveness of the populations of Giglio and Capraia and considering that of Giannutri in part similar to them.

The results of the present study match in part with these earlier observations. In fact, similarities between lizards from Giglio, Capraia and Giannutri are highlighted by morphological data in males and by DP frequencies in both sexes. The discrepancy consists in the fact that, while Giglio and Capraia belong to a well differentiated mtDNA clade, Giannutri belongs to the Tyrrhenian clade. Moreover, considering nuclear microsatellite markers, while for Giglio and Capraia there is a clear isolation that shows no signs of admixture, specimens from Giannutri seem to be admixed with Giglio, Elba and Cerboli. Therefore, it seems reasonable that the population of Giannutri could represent the result of hybridization/introgression between the ancient Giglio–Capraia clade and the Tyrrhenian clade which may have reached Giannutri with historical introductions or during the LGM, when the island was connected to mainland Tuscany. However, the low sample size of Giannutri used for molecular analysis does not allow conclusions to be drawn on this point. Apart from the case of Giannutri, our results converge in the identification of a clearly phenotypically and genetically distinct lineage of lizards on the islands of Giglio and Capraia. While lepidosis is able to separate the populations belonging to the different clades, SVL does not show a great sign of variation across the sampled localities, and this is in accordance with the low effect of insularity on size variation observed for *P. siculus* by Avramo et al. [45]. Another interesting observation is the DP of the lizards of Capraia where males are mostly “reticulated” and females mostly show the striped “campestris” phenotype (the most common in continental Tuscany and on Elba and Cerboli islands). This sex-related difference could be easily explained, as the DP and coloration of *P. siculus* is known to be sexually dimorphic [46]. On the other hand, we didn’t recognize the same sexual dimorphism in the DP frequencies on the other islands. Moreover, the DP of females in Capraia might reflect some form of introgression from the mainland or nearby islands, but such an event should be detected through mtDNA and microsatellites analyses which is not our case at all. Thus, it is more likely that this may represent a case of convergent evolution of the DP, since striped patterns such as the “campestris” phenotype are very common within the genus *Podarcis* with different species showing similar patterns [47]. However, according to mtDNA, the Giglio–Capraia clade is related to the Adriatic clade (sensu Senczuk et al. [25]), which also share some phenotypic characteristics with these two islands, such as the relatively high frequencies of “reticulated” individuals even if “campestris” is still the most common phenotype within the clade [48]. This could suggest that the different DP frequencies of females observed on Capraia may reflect some dimorphic adaptive process. In fact, context-dependent expression of sexual dimorphism has already been observed in the Tuscan Archipelago for *P. muralis* [49]. One more possible explanation could be the mate-choice effect, but females of *P. muralis* have been observed to not actively choose males based on their color morph [50]. Anyway, further investigation is surely needed to shed light on this case of sexual dimorphism.

Montecristo almost exclusively shows the unique “calabresiae” phenotype and this agrees with the lack of nuclear microsatellites admixture with the other islands. However, this distinctiveness is not supported by morphological and mitochondrial data, and this could be due to the fact that the “calabresiae” phenotype can be considered as a subtype of “campestris”, a phenotype commonly found on islands belonging to the Tyrrhenian clade, the same as Montecristo. Moreover, variations in phenotypic frequencies may occur very rapidly in island-dwelling reptiles [51,52]. In fact, the population of this island was probably founded by a small number of individuals introduced from the mainland [53] and the uniqueness of the “calabresiae” phenotype is probably founder effect induced.

The fact that Giglio and Capraia show a peculiar composition of different DP, compared with the almost monomorphic condition of the other islands, would reinforce the hypothesis of a different origin of the populations of *P. siculus* in the Tuscan Archipelago. In fact, it has already been observed that the distribution of DP phenotypes is generally associated with mitochondrial genetic clades in *P. siculus* [48] and this is in agreement with the mtDNA data.

Therefore, a possible explanation for the high diversity of the Italian wall lizard of the Tuscan Archipelago is a scenario of at least two colonization events of the Archipelago. Partially convergent evidence from molecular data, morphology and DPs would suggest an independent origin of the Giglio–Capraia mtDNA clade with respect to the populations of Elba, Cerboli, Montecristo and Giannutri. Particularly relevant is the high genetic distances observed for cytb between this clade and the populations of the other islands and of mainland Tuscany (8.9–9.1% for Capraia and 8.2–8.8% for Giglio). This level of genetic distance is even higher than those observed between some other well established *Podarcis* species (3.3% between *P. raffonei* and *P. waglerianus*; 7.75% between *P. peloponnesiaca* and *P. erhardii* [19,54]). Following the proposed evolutionary rates for cytb in *Podarcis* (1.45–1.59% Myr; [55]), the divergence time from the closest continental clade ranges approximately from 5.04 to 4.59 Myr for Giglio and from 5.17 to 4.72 Myr for Capraia. Therefore, according to these evolutionary rates, the ancient Giglio–Capraia clade would have colonized these islands in the early Pliocene. The high divergence between Giglio and Capraia (5.4%) also supports an ancient insular origin of this clade.

A similar pattern of ancient colonization of an archipelago may recall that of *P. latastei* from the Western Pontine islands which was considered a subspecies of *P. siculus* and recently raised to the species rank [26,27,28]. The fact that the Giglio–Capraia clade is present in two very distant islands could suggest that other islands were also populated in the past by this ancient clade and that it subsequently became extinct and replaced by other *Podarcis* clades. This would strengthen the hypothesis that Giannutri may actually also belong to the ancient Giglio–Capraia clade [17,44] and that introgression/hybridization with the Tyrrhenian clade may have occurred there. Furthermore, differential patterns of colonization have already been observed in several archipelagos for reptiles [56] and, in particular, in the Tuscan Archipelago for *P. muralis* with an ancient clade surviving on Elba, Pianosa and Palmaiola islands and some nearby islets [57]. It is worth noting that the estimated divergence time for these island populations of *P. muralis* is about 2–3 Myr. This time is much more recent than the estimated time of divergence of the insular Giglio–Capraia mtDNA clade of *P. siculus*. Therefore, the most ancient colonization of the islands of the Tuscan Archipelago is that of *P. siculus* and not that of *P. muralis* as previously believed [58].

Given their similarity with the continental populations, it is likely that the presence of *P. siculus* on the islands of Cerboli and Montecristo is the result of historical introductions, as already observed by Capula [53] for Elba Island. This may raise some concerns about the conservation of the endemic Giglio–Capraia lineage which may suffer from genetic introgression from accidental introductions of new lizards. This is probably what happened in Giannutri which belongs to the Tyrrhenian clade but also shows signs of admixture with Giglio and some distinctive morphological and phenotypic characteristics typical of the Giglio–Capraia clade.

## 5. Conclusions

In conclusion, our results shed light on the origin of the Italian wall lizard populations in the Tuscan Archipelago. The remarkable high morphological and genetic diversity found in the Archipelago is probably the result of at least two different waves of colonization: Elba, and the populations of Cerboli and Montecristo probably originated from historical introductions of lizards belonging to the Tyrrhenian clade, while the populations of Giglio and Capraia islands are the last surviving populations of an ancient lineage that colonized the Tuscan Archipelago during the Pliocene which shares a common ancestry with the Adriatic clade (currently present in south-eastern Italy). Giannutri perhaps represents an interesting case of hybridization between the Tyrrhenian clade and that of Giglio–Capraia. Based on the high phenotypic and molecular distinctiveness of this ancient clade, these populations should be treated as distinct units deserving conservation and management efforts as well as further investigation to assess their taxonomic status.

## Figures and Tables

**Figure 1 animals-13-00386-f001:**
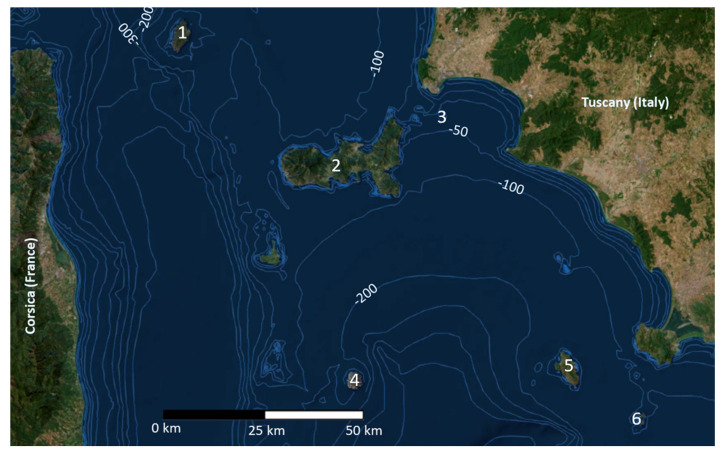
Map of the Tuscan Archipelago. 1: Capraia, 2: Elba, 3: Cerboli, 4: Montecristo, 5: Giglio, 6: Giannutri. Bathymetric map from the open source database of ISPRA (The Italian Institute for Environmental Protection and Research, http://sgi2.isprambiente.it/mapviewer/, accessed on 18 January 2023), modified by Claudia Corti and Andrea Savorelli (Natural History Museum of the University of Florence).

**Figure 2 animals-13-00386-f002:**
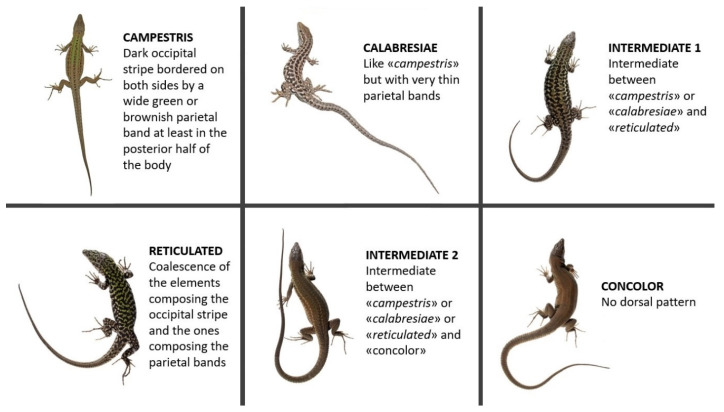
The six main DP categories found in the Tuscan Archipelago. Picture for “campestris” by the anonymous iNaturalist user “gavinh” (link to the observation: https://www.inaturalist.org/observations/27295298; accessed 1 May 2022); picture for “calabresiae” by Mattia Menchetti (link to the observation: www.inaturalist.org/observations/691003; accessed 1 May 2022); pictures for “reticulated”, “concolor”, “intermediate 1” and “intermediate 2” taken by Enrico Lunghi during one of the sampling sessions on Giglio Island (see Lunghi et al. [31] for photographic methods).

**Figure 3 animals-13-00386-f003:**
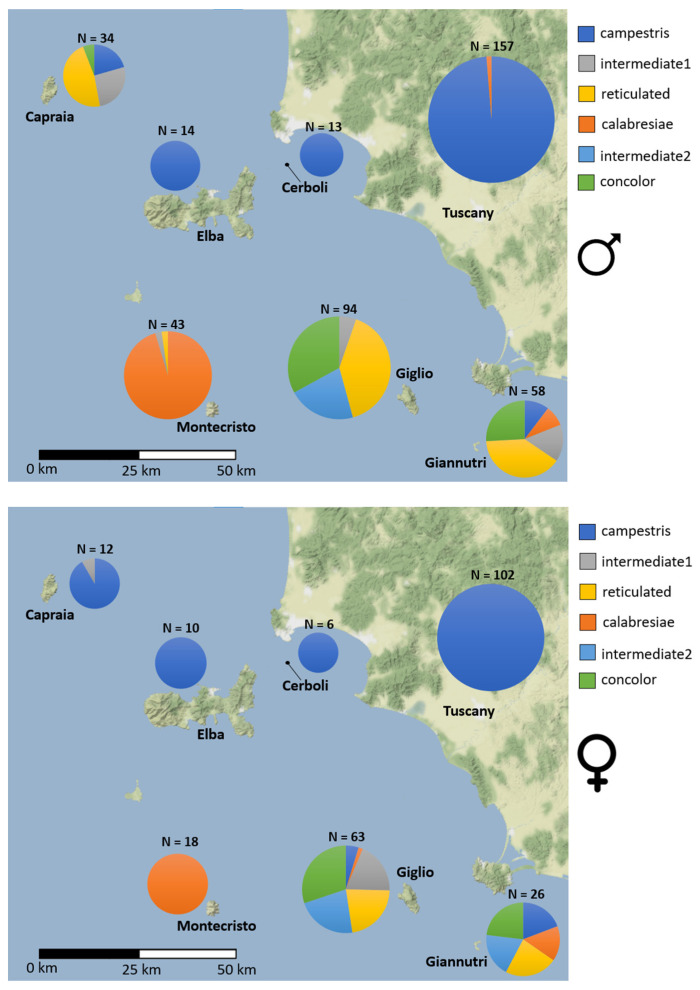
Phenotypic frequencies for males and females in the Tuscan Archipelago and continental Tuscany.

**Figure 4 animals-13-00386-f004:**
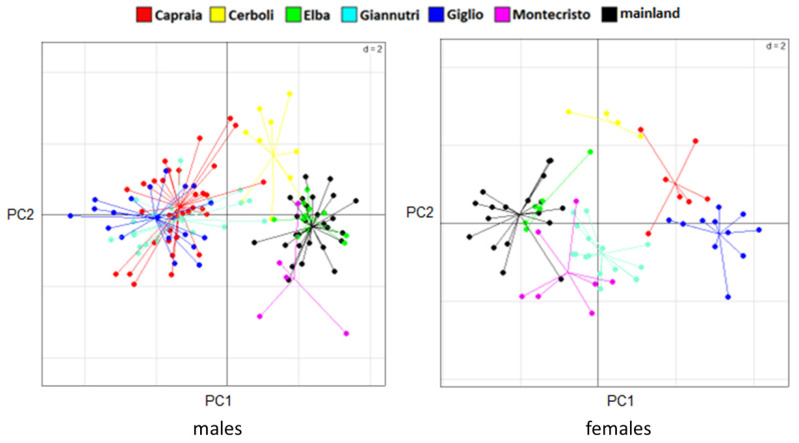
PCA results for males (on the left) and females (on the right). PC1 on the horizontal axes and PC2 on the vertical axes are shown, accounting for 46.83% and 13.12% of the total variance in males, respectively, and 41.23% and 15.36% of the total variance in females, respectively.

**Figure 5 animals-13-00386-f005:**
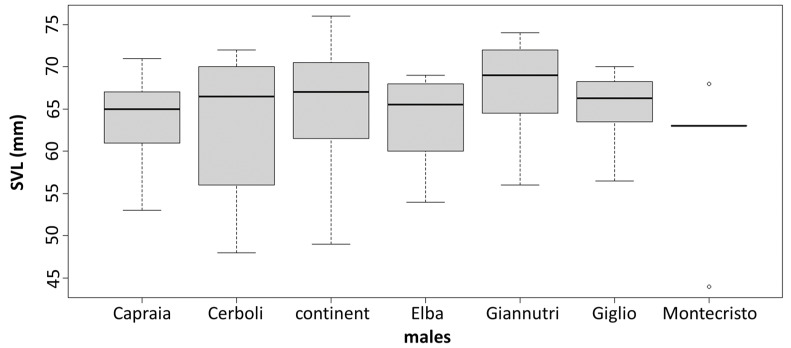
Boxplots for SVL in males and females. There is no significant variation across the sampled localities.

**Figure 6 animals-13-00386-f006:**
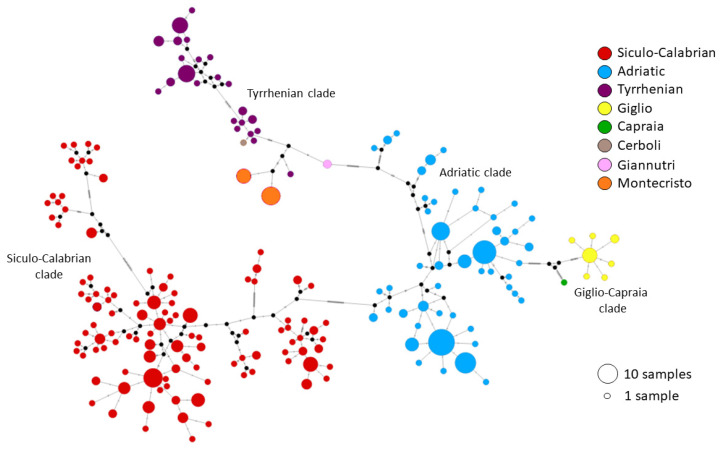
Median joining network built on cytb sequences. Colors represent the three main clades described by Senzcuk et al. [25] and the islands of the Tuscan Archipelago. The Siculo–Calabrian clade includes Sicily, Sardinia, Southern Corsica and the Southern part of Calabria; the Adriatic clade includes Campania, the Adriatic coast, Croatia and the Northern part of Calabria; the Tyrrhenian clade includes Giannutri, Montecristo, Cerboli, Northern Corsica and the Tyrrhenian coast. Giglio and Capraia are shown in yellow and green on the right.

**Figure 7 animals-13-00386-f007:**
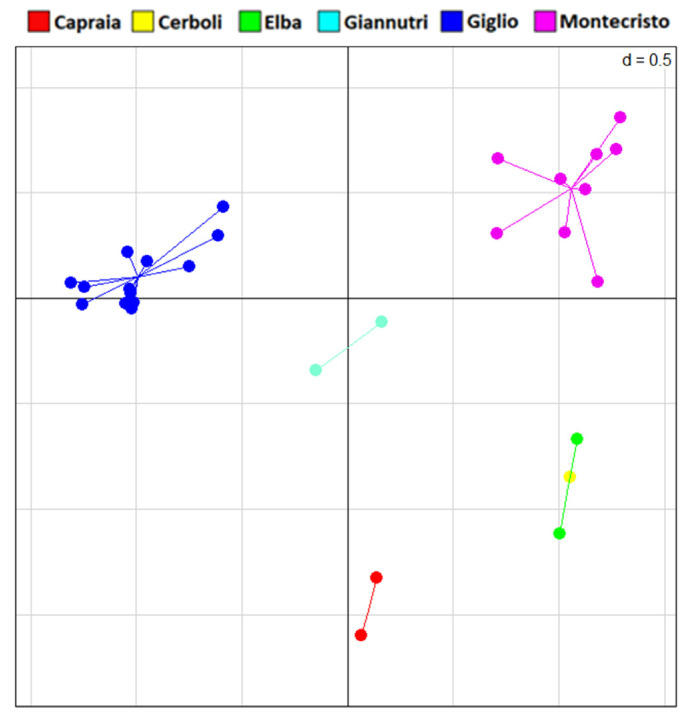
sPCA plot of microsatellites data. Principal component axis 1 and axis 2 are shown here.

**Figure 8 animals-13-00386-f008:**
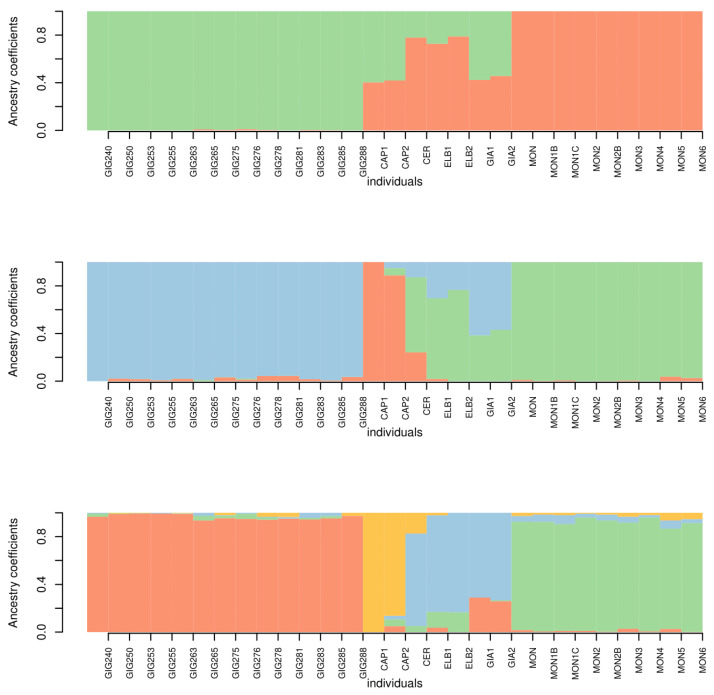
Bar plots for k = 2 to k = 4 (top to bottom). The names of the individuals are labelled below separated by thin white vertical lines (GIG = Giglio, CAP = Capraia, CER = Cerboli, ELB = Elba, GIA = Giannutri, MON = Montecristo).

**Figure 9 animals-13-00386-f009:**
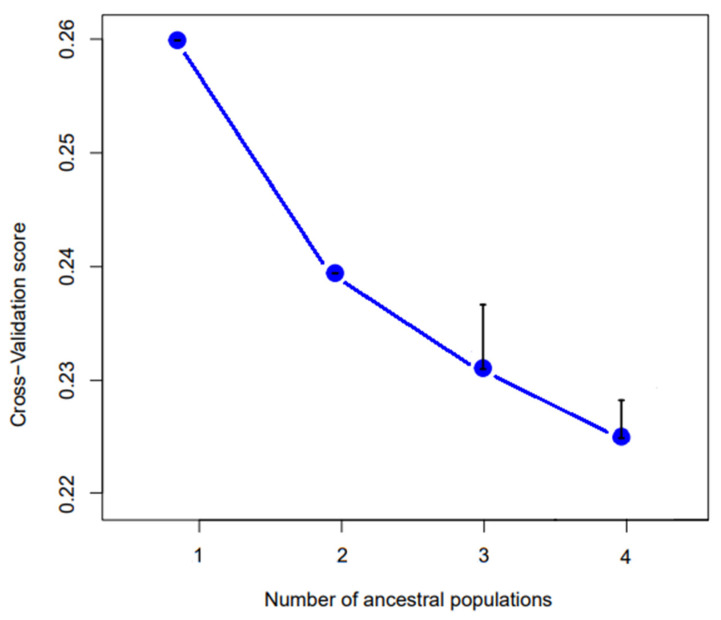
Cross-validation scores for k from 1 to 4. The lower the CV-score, the better fitting is the number of ancestral populations. k = 4 is the best fitting value. Black bars represent the estimated error.

**Table 1 animals-13-00386-t001:** Number of successfully analyzed samples for each island in each analysis. Samples are divided by sex for meristic and DP data (M = males, F = females). This table does not include the cytb sequences coming from GenBank.

Island	mtDNA	Microsatellites	Meristic Data/SVL	Dorsal Pattern
Capraia	1	2	43 M–9 F	34 M–12 F
Cerboli	1	1	16 M–5 F	13 M–6 F
Elba	0	2	17 M–8 F	14 M–10 F
Giannutri	1	2	31 M–19 F	58 M–26 F
Giglio	13	13	26 M–21 F	94 M–63 F
Montecristo	8	9	57 M–11 F	43 M–18 F
mainland	0	0	32 M–19 F	157 M–102 F

**Table 2 animals-13-00386-t002:** MANOVA results. We reported the degrees of freedom (df), Pillai’s Trace, approximate F value (Approx F), number of degrees of freedom in the model (Num df), number of degrees of freedom associated with the model errors (Den df) and *p*-value.

MANOVA (Meristic Data)	df	Pillai’s Trace	Approx F	Num df	Den df	*p*-Value
SEX	1	0.68117	47.003	9	198	<0.05
Locality	6	2.04875	11.695	54	1218	<0.05
Residuals	206					

**Table 3 animals-13-00386-t003:** ANOVA results. We reported the degrees of freedom (df), sum of squares (SS), mean squares (MS), F value and *p*-value.

ANOVA (SVL)	df	SS	MS	F Value	*p*-Value
Sex	1	2828	2827.9	86.357	<0.05
Locality	6	324	54.1	1.652	>0.05
Residuals	206	6746	32.7		

**Table 4 animals-13-00386-t004:** Mean p-distances between *Podarcis siculus* populations of the Tuscan Archipelago and clades (sensu Senczuk et al. [25]) according to our cytb analysis. The Tyrrhenian clade includes all the available sequences from continental Tuscany.

	1	2	3	4	5	6	7	8
1. Giglio	-	-	-	-	-	-	-	-
2. Capraia	5.4%	-	-	-	-	-	-	-
3. Cerboli	8.4%	9.1%	-	-	-	-	-	-
4. Giannutri	8.2%	8.9%	2.2%	-	-	-	-	-
5. Montecristo	8.8%	8.9%	2.4%	1.4%	-	-	-	-
6. Tyrrhenian clade	8.7%	9.1%	0.6%	1.7%	2.3%	-	-	-
7. Siculo-Calabrian clade	8.1%	8.0%	7.5%	7.5%	7.2%	7.7%	-	-
8. Adriatic clade	7.3%	7.5%	5.8%	5.3%	6.5%	5.5%	6.9%	-

## Data Availability

Phenotypic, morphological and microsatellites data are provided as Appendix A. Cytb sequences have been deposited in GenBank, accession numbers from OQ291146 to OQ291157.

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
