# Peer review of "The Intriguing Biogeographic Pattern of the Italian Wall Lizard Podarcis siculus (Squamata: Lacertidae) in the Tuscan Archipelago Reveals the Existence of a New Ancient Insular Clade"

_animals, 2023, doi:10.3390/ani13030386_

Round 1

Reviewer 1 Report

This manuscript tries to find the origin and relationships of the insular populations of Podarcis siculus of the Tuscan archipelago. As with other Mediterranean island populations, the similarities in morphology and coloration do not correspond to the similarities in genetics. In addition, the authors discover that, apparently, two of the populations would have a very ancient origin, related to one of the clades of the mainland, while the rest of the islands would be of more recent introduction from another, closer continental zone. This research opens the door to further research on the colonization process of Podarcis siculus in this archipelago.

Some minor comments:

L. 112. I think Tables S2 and S3 are results and as such it would be better to quote them on line 212, next to the Figure 4 citation.

L. 141 and 147-149 and table S4. The authors say that 9 microsatellites were analysed, but only give details of 8 of them. 

L. 158-160. I think it would be convenient to detail a little more what these samples from GenBank are. I suppose that in GenBank there are more samples of P. siculus, but only those deposited by the authors of the two cited papers have been used. Is this true? In view of the results and the two works cited, I understand that they are continental samples and from some islands, but not the islands of the Tuscan archipelago. Do these GenBank Cytb samples cover the entire distribution area of P. siculus? The authors speak of "the distribution area of P. siculus of our dataset", but I understand as the area of the dataset of this manuscript only the Tuscan Archipelago.

Reviewer 2 Report

The Italian wall lizard is taxonomically complex. Investigating its biogeography, especially on the Tuscan Archipelago is an important endeavor helping to get a better understanding of its evolutionary history and phylogeny. The results impact recognized taxa and conservation efforts and I appreciate the research efforts of the authors!

Overall, the paper is well written. I provided several comments in the materials and methods section. I did not understand the selection of specimens for morphological research and dorsal coloration research and why only specimens from Italian herpetological collections were used (e.g., why not specimens from Mertens’ collection in the SMF?).

In the context of biogeography, maps are very important and the authors should include a scale to visualize the distances of the islands from the continent. Additionally, they could indicate with dashed lines the historical sea levels so one can see the past connections of islands with the continent. The history of island colonization is indeed complicated. I wonder if the authors could include a map with indicated arrows on how islands in time were potentially colonized. Such would be a great support for the current text.

Figure 5 is very important as it visualizes the recognized clades. However, the legend is so small that it is difficult to read. Here, I recommend writing the clade names into the image in proximity to the appropriate network.
